# Gypenoside XVII Reduces Synaptic Glutamate Release and Protects against Excitotoxic Injury in Rats

**DOI:** 10.3390/biom14050589

**Published:** 2024-05-16

**Authors:** Cheng-Wei Lu, Tzu-Yu Lin, Kuan-Ming Chiu, Ming-Yi Lee, Su-Jane Wang

**Affiliations:** 1Department of Anesthesiology, Far-Eastern Memorial Hospital, New Taipei 22060, Taiwan; drluchengwei@gmail.com (C.-W.L.); drlin1971@gmail.com (T.-Y.L.); 2Department of Mechanical Engineering, Yuan Ze University, Taoyuan 32003, Taiwan; 3Division of Cardiovascular Surgery, Cardiovascular Center, Far-Eastern Memorial Hospital, New Taipei 22060, Taiwan; chiu9101018@gmail.com; 4Department of Electrical Engineering, Yuan Ze University, Taoyuan 32003, Taiwan; 5Department of Medical Research, Far-Eastern Memorial Hospital, New Taipei 22060, Taiwan; mingyi.lee@gmail.com; 6School of Medicine, Fu Jen Catholic University, New Taipei 24205, Taiwan; 7Research Center for Chinese Herbal Medicine, College of Human Ecology, Chang Gung University of Science and Technology, Taoyuan 33303, Taiwan

**Keywords:** gypenoside XVII, glutamate release, glutamate excitotoxicity, cerebral blood flow, cortex

## Abstract

Excitotoxicity is a common pathological process in neurological diseases caused by excess glutamate. The purpose of this study was to evaluate the effect of gypenoside XVII (GP-17), a gypenoside monomer, on the glutamatergic system. In vitro, in rat cortical nerve terminals (synaptosomes), GP-17 dose-dependently decreased glutamate release with an IC_50_ value of 16 μM. The removal of extracellular Ca^2+^ or blockade of N-and P/Q-type Ca^2+^ channels and protein kinase A (PKA) abolished the inhibitory effect of GP-17 on glutamate release from cortical synaptosomes. GP-17 also significantly reduced the phosphorylation of PKA, SNAP-25, and synapsin I in cortical synaptosomes. In an in vivo rat model of glutamate excitotoxicity induced by kainic acid (KA), GP-17 pretreatment significantly prevented seizures and rescued neuronal cell injury and glutamate elevation in the cortex. GP-17 pretreatment decreased the expression levels of sodium-coupled neutral amino acid transporter 1, glutamate synthesis enzyme glutaminase and vesicular glutamate transporter 1 but increased the expression level of glutamate metabolism enzyme glutamate dehydrogenase in the cortex of KA-treated rats. In addition, the KA-induced alterations in the N-methyl-D-aspartate receptor subunits GluN2A and GluN2B in the cortex were prevented by GP-17 pretreatment. GP-17 also prevented the KA-induced decrease in cerebral blood flow and arginase II expression. These results suggest that (i) GP-17, through the suppression of N- and P/Q-type Ca^2+^ channels and consequent PKA-mediated SNAP-25 and synapsin I phosphorylation, reduces glutamate exocytosis from cortical synaptosomes; and (ii) GP-17 has a neuroprotective effect on KA-induced glutamate excitotoxicity in rats through regulating synaptic glutamate release and cerebral blood flow.

## 1. Introduction

Glutamate is the predominant excitatory neurotransmitter in the brain that helps to regulate learning, memory, motor function, and mood [1]. However, excessive glutamate in the brain leads to neuronal injury or death, known as glutamate excitotoxicity, which is implicated in many acute and chronic brain diseases, including stroke, traumatic brain injury, seizures, neuropsychiatric disorders, and neurodegenerative diseases such as Alzheimer’s disease (AD), amyotrophic lateral sclerosis (ALS) and Parkinson’s disease (PD) [2,3,4,5]. Inhibition of the glutamatergic system has a neuroprotective effect when brain ischemia, traumatic brain injury, epilepsy, AD, ALS, and mental disease occur [6,7,8]. This phenomenon may be used to prevent and treat these brain diseases.

Phytochemicals derived from medicinal plants offer various medicinal and health benefits to the brain because of their antiapoptotic, antioxidant, and anti-inflammatory properties [9,10], and are important sources of drugs for the treatment of brain disease. Gypenoside XVII (GP-17; molecular formula: C_48_H_82_O_18_; molecular weight: 947.2 g/mol; Figure 1A) is a gypenoside monomer found in medicinal plants such as *Gynostemma pentaphyllum* or *Panax notoginseng*. Several experimental studies have reported various pharmacological properties to GP-17, including immunoregulatory, anti-inflammatory, neuroprotective, cardioprotective, lipid-lowering, and neuroprotective effects [11,12,13,14,15,16]. Regarding its neuroprotective effect, GP-17 attenuated cerebral infarct volume and alleviated neurological dysfunction in a middle cerebral artery occlusion/reperfusion (MCAO/R) rat model [17]. GP-17 also attenuates Aβ-induced oxidative stress, apoptosis, and autophagic cell death in PC12 cells, prevents the formation of Aβ plaques in the hippocampus and cortex, and improves spatial learning and memory deficits in APP/PS1 mice [18,19]. 

Despite the evidence for the neuroprotective effect of GP-17, little is known about its mechanisms, especially the regulation of synaptic glutamate release and the prevention of glutamate excitotoxicity. Therefore, the main aims of the present study were (i) to investigate the effect of GP-17 on glutamate release in an in vitro rat cortical nerve terminals (synaptosomes), a well-established model for studying glutamate release [20], and (ii) to evaluate whether GP-17 has a neuroprotective effect in in vivo rat model of glutamate excitotoxicity via systemic injection of kainic acid (KA), an AMPA/kainate receptor agonist that causes excitotoxicity, which leads to seizures, increased glutamate levels in synapses, and cell death [21,22].

## 2. Materials and Methods

### 2.1. Drugs

GP-17 (purity > 98%) was purchased from ChemFaces (Wuhan, Hubei, China). 4-aminopyridine (4-AP), GF109203X and KT5720 were purchased from Tocris Cookson (Bristol, UK). ω-conotoxin GVIA (ω-CgTX GVIA) and ω-agatoxin IVA (ω-AgTX IVA) were purchased from Alomone lab (Jerusalem, Israel). PKI 14–22, dimethylsulfoxide (DMSO), and all other reagents were purchased from Sigma-Aldrich (St. Louis, MO, USA).

### 2.2. Animals

Sprague Dawley rats (Male, 180–200 g, six weeks old, *n* = 56) were purchased from BioLASCO, (Taipei, Taiwan) and housed in the animal facility of the Fu Jen Catholic University (Taipei, Taiwan). Animals were housed at constant temperature (22 ± 1 °C) and relative humidity (50%) under a regular light–dark schedule (lights 7.00 a.m.–7.00 p.m.). Food and water were freely available. The animal care and experimental procedures complied with the National Institutes of Health guidelines for the care and use of laboratory animals and were approved the Far-Eastern Memorial Hospital Animal Care and Utilization Committee (approval number: IACUC-2022(2)-MOST-02). All efforts were made to minimize pain, discomfort, or suffering of animal and to use only the number of animals necessary to produce reliable results. 

### 2.3. Synaptosomes 

Purified synaptosomes were prepared as previously described [23]. Animals (*n* = 15) were sacrificed by cervical dislocation and the hippocampi were removed as quickly as possible. The tissues were placed in a buffer solution (sucrose 0.32 M, EDTA 1 mM, Tris-HCl 5 mM, pH 7.4) and was homogenized manually with a glass–Teflon tissue homogenizer. To obtain purified synaptosomes the resulting suspension was centrifuged at 3000× *g* for 10 min at 4 °C, and the supernatant was stratified on a Percoll gradient (0, 3, 10, 15, and 23%) and centrifuged at 32,500× *g* for 7 min at 4 °C. The synaptosomal fraction, located between phase 10% and 15% of the gradient, was diluted in HEPES buffer medium (HBM, containing, in mM, 140 NaCl, 5 KCl, 5 NaHCO_3_, 1 MgCl_2_∙6H_2_O, 1.2 Na_2_HPO_4_, 10 glucose, and 10 HEPES) and centrifuged at 27,000× *g* for 10 min at 4 °C to remove Percoll remnants. The obtained pellet was resuspended in HBM and protein was determined according to the method of Bradford (1976) [24], using bovine serum albumin (BSA) as a standard. 0.5 mg of synaptosomal suspension was diluted in HEPES buffer medium and centrifuged at 3000× *g* for 10 min at 4 °C. The supernatants were discarded, and the synaptosomal pellets were stored on ice and used for glutamate release, FM1-43 release and Western blot within 4–6 h.

### 2.4. Evaluation of Glutamate Release and FM1-43 Release in Rat Cortical Synaptosomes

The measurement of released glutamate was performed as described previously [25,26]. Pelleted synaptosomes (0.5 mg/mL) were resuspended in HBM containing 16 μM BSA. The synaptosomal suspension was stirred in a thermostatted cuvette in a Perkin-Elmer LS-55 spectrofluorimeter (PerkinElmer Life and Analytical Sciences, Waltham, MA, USA) at 37 °C, and CaCl_2_ (1.2 mM), NADP^+^ (2 mM) and glutamate dehydrogenase (50 units/mL) were added after 3 min. The depolarizing agent 4-AP (1 mM) was added after another 10 min to induce glutamate release. GP-17 was added to synaptosomes 10 min before 4-AP. Other additions before 4-AP were made as detailed in the figure legends. A standard of exogenous glutamate (5 nmol) was added after another 5 min to calculate the released glutamate. Glutamate release was monitored by measuring the increase in fluorescence (excitation and emission wavelengths of 340 and 460 nm, respectively) caused by NADPH being produced by oxidative deamination of released glutamate by glutamate dehydrogenase. Changes in the released glutamate 5 min after the addition of 4-AP were calculated, and they are expressed herein as nanomoles of glutamate per milligram of synaptosomal protein per 5 min (nmol/mg/5 min). 

FM1-43 is a widely used dye to monitor exocytosis at presynaptic terminals [27]. FM1-43 is a lipophilic but membrane-impermeable fluorescent styryl dye. Thus, when depolarization of nerve terminals results in endocytosis of FM1-43, and during subsequent exocytosis, accumulated dye is released to the extracellular medium, accompanied by a decrease in fluorescence [28]. Briefly, synaptosomes (0.5 mg/mL) were incubated in HBM containing CaCl_2_ (1.2 mM) for 2 min at 37 °C. FM1-43 (0.1 mM) was added for 1 min and then KCl (30 mM) was added for 3 min to load FM1-43. Synaptosomes were rinsed in HBM (2 × 1 min) to remove noninternalized FM1-43. Synaptosomes were then resuspended in HBM containing CaCl_2_ (1.2 mM), and incubated in a thermostatted cuvette in a Perkin-Elmer LS-55 spectrofluorimeter (PerkinElmer Life and Analytical Sciences, Waltham, MA, USA) at 37 °C. Release of loaded FM1-43 was induced by the addition of 1 mM 4-AP, and measured as the decrease in fluorescence upon release of the dye into solution (excitation 488 nm, emission 550 nm). 

### 2.5. KA-Induced Excitotoxicity Animal Model and Drug Treatment

Excitotoxicity was produced in rats that were injected intraperitoneally (i.p.) with KA (15 mg/kg). GP-17 (30 mg/kg) was dissolved in a saline solution containing 1% DMSO and was administered (i.p.) 30 min before KA (15 mg/kg in 0.9% NaCl, pH 7.0, i.p.) injection. The rats were divided into the following four groups: control group (i.p. injection of 1% DMSO; *n* = 10), KA group (only i.p. injection of KA; *n* = 10), GP-17 + KA group (i.p. injection of GP-17 and KA administration; *n* = 10), and GP-17 group (only i.p. injection of GP-17; *n* = 3). The dose and schedule of administration were chosen based on our pilot study and previous studies [29,30,31]. Additionally, seizure behavior was assessed for 3 h after KA administration using the established five stages of seizure activity in the Racine Scale (Racine, 1972) [32]: (1) Mouth and facial movements (orofacial movements), (2) Head nodding (head myoclonus and/or severe orofacial movements), (3) Forelimb clonus, (4) Forelimb clonus with rearing and, (5) Rearing, jumping, and falling with loss of postural control. 

### 2.6. Histological Analysis of Neuronal Degeneration by Fluoro-Jade B (FJB) Staining

FJB staining was used to analyze survived or degenerated neurons in brain sections from rats (*n* = 5 rats per group), as described previously [33,34]. Briefly, three days following KA administration the rats were deeply anesthetized using 3% isoflurane and transcardially perfused with 0.9% saline for 5 min followed by 4% paraformaldehyde for 20 min. The brains were removed and post-fixed in paraformaldehyde overnight at 4 °C. The brains were then removed from the paraformaldehyde solution and transferred to 15% sucrose solution overnight at 4 °C and then followed by 30% sucrose solution for 7 days at 4 °C. Rat brains were sectioned at Bregma levels (−4.7 to −5.0; +4.2 to +3.7) to obtain 25 μm coronal sections using a cryostat. For FJB staining, the sections were mounted onto gelatine-coated slides. Slide mounted sections were airdried for 30 min at 37 °C, immersed in a solution of 1% sodium hydroxide in 80% ethanol for 5 min and then transferred to 70% ethanol for 2 min followed by distilled water for 2 min. Sections were then added to a solution of 0.06% potassium permanganate for 15 min on a rocker, rinsed in distilled water for 1 min then transferred to the FJB Solution (0.0001% FJB solution dissolved in 0.1% acetic acid) for 20 min, and nuclear staining with 4′,6-diamidino-2-phenylindole (DAPI; 1 μg/mL; Sigma-Aldrich, MO, USA). After the staining, slides were rinsed in distilled water (3 × 1 min), dried at 37 °C (20 min), cleared in xylene (2 × 1 min) and cover-slipped with mounting medium DPX (Sigma-Aldrich). Finally, the cortex region was visualized under 100× magnification using an upright fluorescence microscope (Zeiss Axioskop 40, Goettingen, Germany). Quantification of neuronal degeneration by FJB staining was measured in an area of 255 × 255 μm of the prefrontal and entorhinal cortex in three randomly chosen sections from each animal and averaged for each animals using ImageJ image analysis software (Version 1.53k, NIH Image, National Institutes of Health, Bethesda, MD, USA) by an examiner blinded to the experimental conditions. The results are expressed as the mean ± SEM of labelled cells per 0.1 mm^2^.

### 2.7. High-Performance Liquid Chromatography (HPLC) Assay of Glutamate and γ-Aminobutyric Acid (GABA) Concentrations in the Cortex

The glutamate levels in the cortex were measured using HPLC as described previously [35]. Briefly, rats (*n* = 6 rats per group) were sacrificed by decapitation, and the brains were quickly removed and dissected on ice, and the cortex and hippocampus were stored at −80 °C. The frozen cortex was homogenized in HEPES buffer medium and centrifuged at 15,000× *g* for 10 min at 4 °C. The supernatant (10 μL) was filtered through a 0.22 µm membrane filter and injected into an HPLC instrument (HTEC-600, Eicom, Kyoto, Japan). The glutamate and GABA concentrations were determined using peak areas with an external standard method and is expressed herein as ng/mg protein.

### 2.8. Cerebral Blood Flow Monitoring by Laser Speckle Imaging System

Cerebral blood flow in rats was monitored using a Laser Speckle Imaging System (RFLSI III, RWD, China) [36]. Briefly, the rats (*n* = 3–4 rats per group) were anesthetized by 3% isoflurane and placed prone in a stereotaxic head frame. Then, a midline incision was made over the skull to expose the calvaria. The exposure area was kept clean and dry using a tampon during image collection. Then, skull is irradiated by a laser, the high-resolution blood flow speckle image is recorded by the CMSO camera. Real, pcolor, and speckle pictures are shown in the laser speckle imaging system. The cerebral blood flow was continuously measured for 15 s at the following settings: observation height, 25 cm; laser irradiation area, 1.7 cm; image matrix, 2064 × 1544. The cerebral blood flow data were expressed in perfusion unit.

### 2.9. Protein Isolation and Western Blotting

The synaptosomes (*n* = 5 rats per group) or frozen cortex (*n* = 5 rats per group) was homogenized in 2% SDS containing 1 mM PMSF, 1 mM Na_2_VO_4_, 20 mM NaF, and a mixture of phosphatase-proteinase inhibitors (Sigma Aldrich) using ultrasonic homogenizers. After denaturation at 95 °C for 10 min, insoluble debris was removed by centrifugation at 10,000× *g* for 10 min at 4 °C. The protein concentration in the supernatants was determined using a Bradford assay kit (Bio-Rad Laboratories, Hercules, CA, USA). Equal aliquots of protein (20 μg/lane) from each sample were separated via SDS–PAGE (10–15% gel) and transferred to polyvinylidene fluoride (PVDF) membranes (Thermo Scientific, Wilmington, MA, USA). The membranes were treated with 5% bovine serum albumin in Tris-buffered saline (TBS) supplemented with 0.1% Tween-20 (TBST) for 40 min at room temperature and then probed with specific primary antibodies overnight at 4 °C. The primary antibodies were used as follows: protein kinase A (PKA, 1:2000, Cell Signaling, Beverly, MA, USA), PKA pThr197 (1:2000, Cell Signaling, Beverly, MA, USA), synapsin I (1:50,000, Cell Signaling, Beverly, MA, USA), synapsin I pSer9 (1:700, Cell Signaling, Beverly, MA, USA), SNAP-25 (1:50,000; Abcam, Cambridge, UK), SNAP-25 pThr138 (1: 500; Abcam, Cambridge, UK), sodium-coupled neutral amino acid transporter 1 (SNAT1) (1:1000; Invitrogen, Waltham, MA, USA), glutaminase (1:10,000; Invitrogen, Waltham, MA, USA), glutamate dehydrogenase (GDH) (1:30,000; Invitrogen, Waltham, MA, USA), vesicular glutamate transporter 1 (VGLUT1) (1:9000, Gentex, Zeeland, MI, USA), GluN2A (1:1000, Cell Signaling, Beverly, MA, USA), GluN2B (1:1000, Cell Signaling, Beverly, MA, USA), GluN1 (1:1000, Cell Signaling, Beverly, MA, USA), arginase II (ArgII) (1:1000, Cell Signaling, Beverly, MA, USA), and β-actin (1:10,000, Cell Signaling, Beverly, MA, USA). After washing with TBST 5 times for 6 min, the membranes were incubated for 2 h at 25 °C with horseradish peroxidase-conjugated secondary antibodies (1:2000, Gentex, Zeeland, MI, USA). Expressed proteins of interest were visualized by enhanced chemiluminescence solution (Amersham Biosciences Corp., Amersham, Buckinghamshire, UK). The films were scanned using a scanner and quantified with ImageJ software (Version 1.53k, NIH Image, National Institutes of Health, Bethesda, MD, USA), after which the protein/β-actin ratio was calculated [35,37]. The measurements of β-actin were conducted on sister blots, with each blot loaded with 30 μg of protein per lane. 

### 2.10. Statistical Analysis

A Shapiro–Wilk test was performed to assess the normality of the dataset. GraphPad Prism 8 (San Diego, CA, USA) was utilized for data analysis and for drawing graphs. Analysis of variance was performed using ANOVA followed by Tukey–Kramer multiple-comparisons test. Results were considered significant at *p* < 0.05. 

## 3. Results

### 3.1. GP-17 Reduces Ca^2+^-Dependent Glutamate Release Evoked by 4-AP in Rat Cerebral Cortex Glutamatergic Nerve Terminals 

We were interested in determining whether GP-17 regulates the release of glutamate from cortical synaptosomes in rats. Synaptosomes were pretreated with 10 μM GP-17 for 10 min before 4-aminopyridine addition, as shown in Figure 1B. GP-17 (10 μM) did not regulate the basal release of glutamate, but reduced the 1 mM 4-AP-evoked glutamate release from cerebrocortical synaptosomes [t(20) = 28.1; *p* < 0.0001]. GP-17 inhibited the release of glutamate in a concentration-dependent manner (3–50 μM) and was maximally active at 30 μM (Figure 1C). Similar results were observed for the release of glutamate evoked by 15 mM KCl (*p* < 0.0001; Figure 1C). Given the robust depression of glutamate release seen with 10 μM GP-17, this concentration was used in subsequent experiments to evaluate the mechanisms underlying the ability of GP-17 to reduce glutamate release.

In Figure 1D, the release of glutamate evoked by 4-AP was largely reduced by the addition of 0.3 mM EGTA to synaptosomes (incubated in the absence of external Ca^2+^) [F(2,12) = 2732; *p* < 0.0001]. This Ca^2+^-independent glutamate release was not affected by 10 μM GP-17 (*p* = 0.99; Figure 1D), suggesting that the effect of GP-17 is related to a decrease in Ca^2+^ influx from external Ca^2+^. Moreover, 4-AP-evoked glutamate release from rat cerebrocortical synaptosomes is mediated by N- and P/Q-type Ca^2+^ channel opening [38]. To confirm the involvement of presynaptic N- and P/Q-type Ca^2+^ Channels in glutamate release, experiments were carried out to verify whether the N- and P/Q-type Ca^2+^ channel antagonists ω-CgTX GVIA and ω-AgTX IVA can antagonize the inhibitory effect elicited by GP-17. As shown in Figure 1D, 4-AP-evoked glutamate release was significantly reduced by ω-CgTX GVIA (2 μM) and ω-AgTX IVA (0.5 μM) [F(2,12) = 1848; *p* < 0.0001]. When ω-CgTX GVIA (2 μM) and ω-AgTX IVA (0.5 μM) were present, GP-17 did not affect the glutamate release evoked by 4-AP (*p* = 0.89). The lack of further inhibitory effects of GP-17 and the combined application of ω-CgTX GVIA and ω-AgTX IVA on 4-AP-evoked glutamate release suggested that there is a preferential interaction between the pathway mediated by GP-17 and N- and P/Q-type Ca^2+^ channels. Furthermore, an FM1-43 dye-based exocytosis assay showed that GP-17 inhibited the FM1-43 release evoked by 4-AP [t(8) = 16.4; *p* < 0.0001; Figure 1E]. These results suggest that the GP-17-mediated inhibition of 4-AP-evoked glutamate release is mediated by a reduction in Ca^2+^-dependent exocytotic release. The dose of administration of ω-CgTX GVIA and ω-AgTX IVA were chosen based on previous studies [25,26]. 

### 3.2. Suppression of the Protein Kinase A Pathway Is Involved in the GP-17-Mediated Inhibition of Glutamate Release from the Cerebrocortical Synaptosomes of Rats

The release of glutamate evoked by 4-AP in nerve terminals is regulated by numerous protein kinases, particularly protein kinase A (PKA) and protein kinase C (PKC) [39,40]. Therefore, we tested the effect of the PKC inhibitor GF109203X or the PKA inhibitor KT5720 on the inhibitory effect elicited by GP-17. As shown in Figure 2A, synaptosomes were pretreated with 10 μM GF109203X or 10 μM KT5720 for 30 min before 4-AP addition. GF109203X or KT5720 decreased 4-AP-evoked glutamate release [GF109203X, F(2,12) = 970.4, *p* < 0.0001; KT5720, F(2,12) = 195.5; *p* < 0.0001]. When GF109203X and GP-17 were applied simultaneously, the inhibitory effect of glutamate release following 4-AP depolarization was significantly different from the inhibitory effect of GF109203X alone (*p* < 0.0001). However, GP-17 did not affect 4-AP-evoked glutamate release in the presence of KT5720 (*p* = 0.99). Similar results were also obtained with another PKA inhibitor, PKA inhibitor peptide 14–22 (PKI 14–22). As with KT5720, 1 μM PKI 14–22 decreased the glutamate release evoked by 4-AP [F(2,12) = 86.9; *p* < 0.0001]. When PKI 14–22 and GP-17 were applied simultaneously, the inhibitory effect of glutamate release evoked by 4-AP was not significantly different from the inhibitory effect of PKI 14–22 alone (*p* = 0.96). These results indicate the involvement of PKA suppression in the GP-17-induced inhibition of 4-AP-evoked glutamate exocytosis. To confirm this hypothesis, the effect of GP-17 on the phosphorylation of PKA was analyzed by Western blotting, as shown in Figure 2B–D. Treatment with 1 mM 4-AP markedly increased the phosphorylation of PKA at Thr197 in synaptosomes [F(2,12) = 123.1; *p* < 0.0001]. When synaptosomes were pretreated with 10 μM GP-17 for 10 min before 4-aminopyridine addition, a significant decrease in 4-AP-induced PKA (Thr197) phosphorylation was observed (*p* < 0.0001). No significant differences in PKA expression levels were observed among the groups [F(2,12) = 0.2; *p* = 0.82; Figure 2C]. 

In addition, PKA has been shown to phosphorylate synapsin I and SNAP-25, and this phosphorylation is involved in controlling the size of the releasable vesicle pools [41]. The effect of GP-17 on the phosphorylation of synapsin I and SNAP-25 was also assessed, as shown in Figure 2E–H. The phosphorylation of synapsin I Ser9 or SNAP-25 Thr138 was increased by 1 mM 4-AP [synapsin I Ser9, F(2,12) = 464.5, *p* < 0.0001, Figure 2G; SNAP-25 Thr138, F(2,12) = 356.1, *p* < 0.0001, Figure 2I], and this phenomenon was reduced in the presence of GP-17 (*p* < 0.0001). No significant differences in synapsin I or SNAP-25 expression were observed among the groups [synapsin I, F(2,12) = 0.04, *p* = 0.96, Figure 2F; SNAP-25, F(2,12) = 0.02, *p* = 0.98, Figure 2H].

### 3.3. GP-17 Prevents KA-Induced Excitotoxicity in Rats

The model of excitotoxicity in rats can be generated by i.p. injection of KA [21,42]. As shown in Table 1, the number of rats with seizures was 20/20 in the KA group, and 0/14 in the 30 mg/kg Gp-17 + KA group. GP-17 administration (i.p.) at 30 mg/kg 30 min before KA administration had a significant effect on the seizure score [F(2,41) = 136.5; *p* < 0.0001], but other doses (10 mg/kg) had no significant effect on the seizure onset time or seizure score compared with those in the KA group (*p* > 0.05). In addition, we used FJB and DAPI staining of brain sections in situ to evaluate the extent of KA-induced neuronal injury in the cortex. As shown in Figure 3A, FJB labeling was not observed in the prefrontal or entorhinal cortex of the control rats. In KA-treated rats, FJB fluorescence intensity was increased in neurons of the prefrontal and entorhinal cortex after 72 h, indicating neural injury. Quantification of FJB labeling in the prefrontal and entorhinal cortex is shown in Figure 3B,C, which shows that the number of FJB-positive cells in the KA group was greater than that in the control group [prefrontal cortex, F(3,14) = 978.3, *p* < 0.0001; entorhinal cortex, F(3,14) = 152.1, *p* < 0.0001]. However, the number of FJB-positive cells in the GP-17 + KA group was lower than that in the KA group (*p* < 0.0001). The GP-17-only group showed no obvious neuronal degeneration in the prefrontal and entorhinal cortex compared with the control group (*p* = 1; *n* = 3 per group; Figure 2A–C).

### 3.4. GP-17 Decreases Glutamate Concentrations in the Cortex of KA-Treated Rats

KA-induced seizures and neuronal death are linked to excess glutamate [21,43]. As shown in Figure 4A, the levels of glutamate in the cortex were significantly greater in the KA-treated group than in the control group after 72 h [F(2, 15) = 45.3; *p* < 0.0001]. However, the level of GABA in the cortex did not significantly differ between the control group and the KA-treated group [F(2, 15) = 0.49; *p* = 0.62; Figure 4B]. Therefore, the glutamate/GABA ratio in the cortex was greater in the KA group than in the control group [F(2,15) = 31.8; *p* < 0.0001; Figure 4C]. In addition, the glutamate concentration and glutamate/GABA ratio in the cortex were significantly lower in the 30 mg/kg GP-17 + KA group than in the KA-treated group (*p* < 0.0001; Figure 4A,C).

### 3.5. GP-17 Decreases the Protein Levels of SNAT1, Glutaminase and VGLUT1 but Increases the Protein Level of GDH in the Cortex of KA-Treated Rats

We analyzed the protein levels of the glutamine transporter SNAT1, glutamate-generating enzyme glutaminase, glutamate-metabolizing enzyme GDH and glutamate transporter VGLUT1 in the cortex, which are related to KA-induced glutamate elevation in the brain [21,35,42,44]. As shown in Figure 5A–E, after 72 h, the protein levels of SNAT1, glutaminase, and VGLUT1 were significantly greater [SNAT1, F(2,12) = 1268, *p* < 0.0001; glutaminase, F(2,12) = 1905, *p* < 0.0001; VGLUT1, F(2,12) = 2743, *p* < 0.0001], while GDH protein expression was significantly lower in the cortex of the KA group than in that of the control group [F(2,12) = 262.2; *p* < 0.0001]. In the cortex of the GP-17 + KA group, the protein levels of SNAT1, glutaminase and VGluT1 decreased (*p* < 0.001), while the protein level of GDH increased compared to those in the KA group (*p* < 0.001; Figure 5).

### 3.6. GP-17 Alters the Protein Expression of the N-methyl-D-aspartate (NMDA) Receptor Subunits GluN2A and GluN2B in the Cortex of KA-Treated Rats

Alterations in the expression of the NMDA receptor subunits GluN2A and GluN2B also occurred in KA-induced excitotoxic brain injury [29,35]. The protein levels ofGluN2A and GluN2B were analyzed, as shown in Figure 6. The protein level of GluN2A decreased [GluN2A, F(2,12) = 604.8; *p* < 0.0001], while that of GluN2B increased in the cortex of the KA group after 72 h compared with that in the cortex of the control group [GluN2B, F(2,12) = 1950; *p* < 0.0001]. However, the GluN2A protein was increased and the GluN2B protein was decreased in the cortex of the GP-17 + KA group compared to the KA group (*p* < 0.0001). No significant differences in GluN1 expression levels were observed among the groups [F(2,12) = 0.04; *p* = 0.96; Figure 2A,D].

### 3.7. GP-17 Prevents Decreases in Cerebral Blood Flow and ArgII Expression in KA-Treated Rats 

Decreased cerebral blood flow is related to excitotoxic neuronal cell death [5,45]. To further verify the neuroprotective effects of GP-17 on KA-induced excitotoxicity, cerebral blood flow in the whole brain was assessed using a laser speckle imaging system at 72 h after KA injection. As shown in Figure 7A,B, cerebral blood flow in the brain was significantly lower in the KA group than in the control group [F(2,7) = 22.9; *p* < 0.001]. However, cerebral blood flow in the brain was significantly greater in the GP-17 + KA group than in the KA group (*p* = 0.007). In addition, we analyzed the protein levels of ArgII, which regulates cerebral blood flow and plays a protective role in excitotoxicity [46]. As shown in Figure 7C,D, the protein level of ArgII was decreased in the cortex of the KA group after 72 h compared with that in the cortex of the control group [F(2,12) = 821.4; *p* < 0.0001]. However, the protein level of ArgII in the cortex was greater in the GP-17 + KA group than in the KA group (*p* < 0.0001). 

## 4. Discussion

The main finding of the present study was that pretreatment with GP-17, a gypenoside monomer, inhibits 4-AP-evoked glutamate release from rat cerebrocortical synaptosomes and has a neuroprotective effect on KA-induced glutamate excitotoxicity in rats.

Excitotoxicity is a phenomenon of neuronal toxicity that occurs in many pathological conditions and is caused by excess glutamate [2,3]. The regulation of glutamate release is critical for preserving glutamate homeostasis and ensuring neuronal functionality. In the present study, we used nerve terminals isolated from the rat cerebral cortex, a well-documented model for studying glutamate release, and found that GP-17 dose-dependently decreased glutamate release evoked by 1 mM 4-AP, with an IC_50_ of 16 μM. Similarly, the effect of GP-17 on 15 mM KCl-evoked glutamate release, which involves only Ca^2+^ channel activation, was concentration dependent, and the IC_50_ was approximately 20 μM. In addition, in the absence of extracellular Ca^2+^ or blockade of N- and P/Q-type Ca^2+^ channels, GP-17 had no effect on glutamate release induced by 4-AP. These results indicate a close correlation between the inhibition of glutamate release by GP-17 and decreased Ca^2+^ influx through N- and P/Q-type Ca^2+^ channels. However, how GP-17 affects these Ca^2+^ channels is unclear. Whether GP-17 interacts with these Ca^2+^ channels remains to be addressed. 

In nerve terminals, decreased Ca^2+^ influx affects numerous protein kinases, particularly PKA [40,47]. PKA regulates glutamate exocytosis through the phosphorylation of synaptic proteins [41]. For example, PKA phosphorylates SNAP-25 at Thr138, a key protein for the fusion of synaptic vesicles with the plasma membrane during exocytosis, and is involved in controlling the size of releasable vesicle pools [48]. PKA also phosphorylates synapsin I at Ser9, which is a cytoplasmic surface synaptic vesicle protein. Its phosphorylation on Ser9 decreases actin binding, allowing synaptic vesicles to mobilize to the releasable pool [49,50]. In the present study, we found that in the presence of PKA inhibitors, GP-17 did not affect 4-AP-evoked glutamate release from cortical synaptosomes. This finding supports a role for PKA in the GP-17-mediated decrease in glutamate release. In addition, in line with previous studies, 4-AP increased PKA pThr197, SNAP pThr138, and synapsin I pSer9 levels in cerebrocortical synaptosomes [51,52], an effect possibly associated with an increase in vesicle trafficking. However, GP-17 significantly reduced the 4-AP-induced increase in the levels of PKA pThr197, SNAP pThr138, and synapsin I pSer9. These data indicate that GP-17 inhibits glutamate release in cortical synaptosomes by decreasing the number of readily releasable vesicles via the suppression of PKA, SNAP-25, and synapsin I phosphorylation. In addition, we found that the inhibitory effect of GP-17 on 4-AP-evoked glutamate release was not affected by the PKC inhibitor GF109203X. In the presence of GF109203X, GP-17 still induced a 49% decrease in the glutamate release. The additive relationship between GF109203X and GP-17 indicates that PKC is not involved in the effect of GP-17 on glutamate release. Based on these observations, we propose that GP-17, through the suppression of N- and P/Q-type Ca^2+^ channels and consequent PKA-mediated SNAP-25 and synapsin I phosphorylation, reduces glutamate exocytosis from cortical synaptosomes (Figure 8A).

We also elucidated the effect of GP-17 on neuronal injury in a rat model of excitotoxicity induced by systemic administration of KA, an excitotoxic glutamate analog. Consistent with previous studies [35,53,54], I.P. administration of 15 mg/kg KA induced seizures and neuronal injury in the entorhinal cortex. Pretreatment with 30 mg/kg GP-17 inhibited the convulsive effects of KA by increasing the latency to seizure onset and reducing seizure severity. GP-17 pretreatment also blocked KA-induced neuronal injury, as indicated by an increase in the number of surviving neurons (as indicated by NeuN staining) and a decrease in the number of degenerated neurons (as indicated by FJB staining) in the entorhinal cortex of KA-treated rats. These results suggest that GP-17 can prevent seizures and neuronal injury caused by KA in the rat brain.

KA-induced seizures and neuronal death are linked to excess glutamate, which is likely due to increased glutamate release. This increase is related to the level and activity of SNAT1, glutaminase, GDH, and VGLUT1 [21,35,42,44]. When SNAT1 takes up synaptic glutamine, glutaminase converts glutamine to glutamate within neurons, GDH metabolizes glutamate to α-ketoglutarate, and VGLUT1 transports glutamate into presynaptic vesicles and subsequently releases it into the synaptic cleft [55,56]. In particular, excitotoxicity is directly related to the level of VGluT1 [7,57]. In the present study, a significant increase in glutamate and the protein levels of SNAT1, glutaminase, and VGLUT1 and a marked decrease in the protein level of GDH were detected in the cortex of KA-treated rats. These results suggest that increased glutamate synthesis and release and decreased glutamate metabolism lead to an increase in glutamate in the cortex of KA-treated rats. Additionally, GP-17 pretreatment decreased glutamate, SNAT1, glutaminase and VGLUT1 levels and increased GDH levels in the cortex of KA-treated rats. These findings suggested that GP-17 preserves the normal synthesis, metabolism, and synaptic release of glutamate, a likely explanation for the decreased glutamate level in the cortex of KA-treated rats (Figure 8B). These effects might also be a mechanism underlying the neuroprotective effect of GP-17 against KA-induced excitotoxic injury.

Excess glutamate induced by KA overactivates glutamate receptors at excitatory synapses, especially NMDA receptors, which are highly responsive to Ca^2+^ and are involved in excitotoxicity [44,58]. NMDA receptors consist of two obligatory GluN1 subunits and various combinations of the same or different GluN2 (GluN2A-2D) and GluN3 subunits [59]. Among the NMDA receptor subunits, the GluN2 subunit mainly determines NMDA receptor function, particularly the GluN2A and GluN2B subunits, which are highly expressed in excitatory neurons and have differential roles in mediating excitotoxic neuronal death [60]. For example, the activation of GluN2A-containing NMDA receptors promotes neuronal survival and exerts a neuroprotective effect, whereas the activation of GluN2B-containing receptors causes an excessive increase in intracellular Ca^2+^, leading to neuronal death [61]. Thus, decreased expression of the GluN2A subunit subsequently allows an increased influx of Ca^2+^ upon glutamate stimulation and thus, increased excitotoxicity [58,62]. In the present study, a significant decrease in the protein level of GluN2A and a marked increase in the protein level of GluN2B were detected in the cortex of KA-treated rats. These results were consistent with those of previous studies [29,35,54]. Additionally, these alterations in GluN2A and GluN2B protein levels were reversed by pretreatment with GP-17. Thus, our data suggest that in addition to decreasing NMDA receptor activation, alterations in NMDA receptor composition, particularly increases in proportion to the amount of GluN2A-containing NMDA receptors, are related to the ability of GP-17 to prevent cortical neuronal death in KA-treated rats (Figure 8B).

In addition, cerebral blood flow homeostasis is necessary for the normal functioning of neurons, and insufficient cerebral blood flow renders neurons vulnerable to excitotoxic damage [5,45,63]. In the present study, we found a marked decrease in cerebral blood flow in KA-treated rats. This impaired cerebral blood flow was also reversed by GP-17 pretreatment. These results suggested that GP-17 preserves normal cerebral blood flow, thus contributing to the decrease in neuronal injury in KA-treated rats. In addition, we also detected a decrease in ArgII expression in the cortex of KA rats. On the other hand, GP-17 pretreatment upregulated the expression of ArgII in the cortex of KA-treated rats. ArgII is an important enzyme that regulates nitric oxide (NO) production and cerebral blood flow and protects the brain from ischemic and excitotoxic damage [46]. In particular, the deletion of ArgII not only reduces cerebral blood flow but also increases the availability of L-arginine as a substrate for nitric oxide synthase (NOS) and increases NO generation, which leads to reactive oxygen/nitrogen species generation and augmented NMDA-induced toxicity [46,64]. Thus, GP-17 was inferred to have a neuroprotective effect on KA-induced cortical neuronal cell injury, and this effect may also be related to the maintenance of ArgII expression and the control of cerebral blood flow. However, the detailed mechanism by which GP-17 modulates ArgII and cerebral blood flow was not elucidated in this study and should be investigated further.

Studies in laboratory animals indicate that GP-17 has neuroprotective effects [17,18,65]. However, the mechanisms underlying the neuroprotective effects of GP-17 are still under investigation. The results described in this report demonstrate that GP-17 inhibits glutamate release from cerebrocortical synaptosomes via the suppression of voltage-dependent Ca^2+^ channel activity and attenuates glutamate concentration by decreasing the synaptic release of glutamate in the cortex of KA-treated rats. Since some neuroprotective agents, including plant-derived components, have been revealed to diminish the synaptic release of glutamate in brain tissues [25,26,66], decreasing synaptic glutamate release may explain, in part, the neuroprotective mechanism of GP-17. In addition to the regulation of synaptic glutamate release, the observed anti-excitotoxic effect of GP-17 might also be related to the regulation of ArgII and cerebral blood flow. The relevance of our findings to in vivo clinical situations remains to be determined. In addition, the dose of GP-17 used to prevent neuronal damage in our in vivo study was 30 mg/kg. Consistent with our study, studies have revealed that the effect of GP-17 occurs at 10–50 mg/kg in vivo [17,65]. Other gypenosides administered at doses of 100–200 mg/kg can attenuate depression- or anxiety-like behavior of the mice [67] and improve memory impairment in rats [68].

## 5. Conclusions

Using rat cortical synaptosomes and KA-induced glutamate excitotoxicity in rats, this study suggested that GP-17 inhibits synaptic glutamate release, excitotoxicity and regulates cerebral blood flow. These findings provide a better understanding of the beneficial effects of GP-17 in the central nervous system and insights into its clinical use for neurological disorders involving glutamate excitotoxicity. 

## Figures and Tables

**Figure 1 biomolecules-14-00589-f001:**
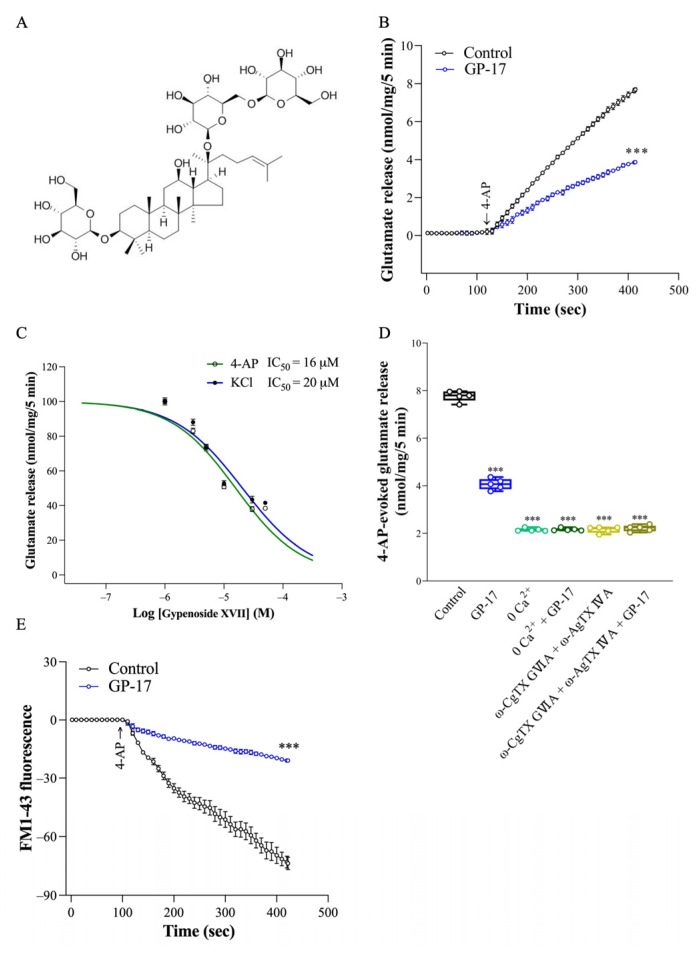
Effect of GP-17 on the 4-AP-evoked glutamate release from synaptosomes isolated from the cerebral cortex of adult rats. (**A**) Chemical structure of GP-17. (**B**) Glutamate release was measured under control conditions or in the presence of 10 μM GP-17. ***, *p* < 0.0001, two-tailed Student’s *t* test. (**C**) Dose-response curve for GP-17 inhibition of 1 mM 4-AP- and 15 mM KCl-evoked glutamate release. (**D**) Glutamate release by 1 mM 4-AP in the absence and presence of 10 μM GP-17 and absence and presence of omitting CaCl_2_ and adding 0.3 mM EGTA, 2 μM ω-CgTX GVIA, or 0.5 μM ω-AgTX IVA. ***, *p* < 0.0001 versus control group, One-way ANOVA with Tukey posthoc test. (**E**) FM1–43 release by 1 mM 4-AP in the absence (control) or presence of 10 μM GP-17. ***, *p* < 0.0001, two-tailed Student’s *t* test. GP-17 was added 10 min before depolarization and ω-CgTX GVIA or ω-AgTX IVA was added 20 min before this. *n* = 5 rats per group.

**Figure 2 biomolecules-14-00589-f002:**
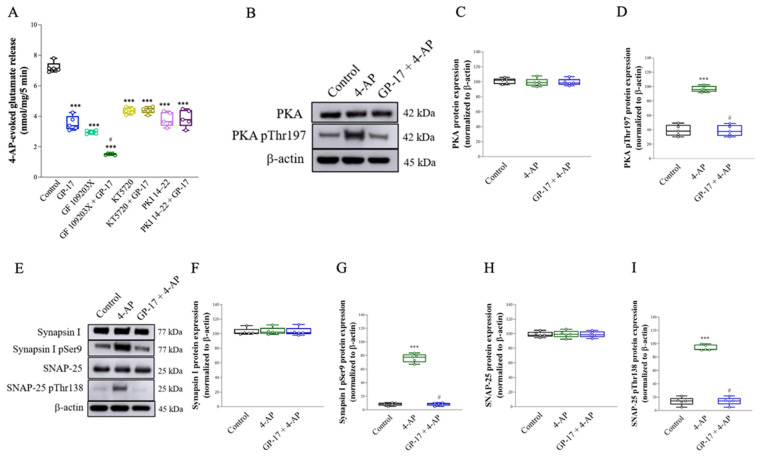
Involvement of PKA suppression in the inhibition caused by GP-17 on glutamate release from cerebrocortical synaptosomes. (**A**) Glutamate release by 1 mM 4-AP in the absence and presence of 10 μM GP-17 and absence and presence of 10 μM GF109203X, 10 μM KT5720, or 1 μM PKI 14–22. (**B**,**C**) Immunoblot results of PKA, PKA pThr197, synapsin I, synapsin I pSer9, SNAP-25, and SNAP-25 pThr138, and β-actin proteins in the synaptosomes from different groups. (**C**,**D**,**F**–**I**). The relative protein levels were quantified. ***, *p* < 0.001 versus the control group, #, *p* < 0.05 versus the 4-AP-treated group. One-way ANOVA with Tukey posthoc test. *n* = 5 rats per group. The details of the Western blot original images are in Appendix A.

**Figure 3 biomolecules-14-00589-f003:**
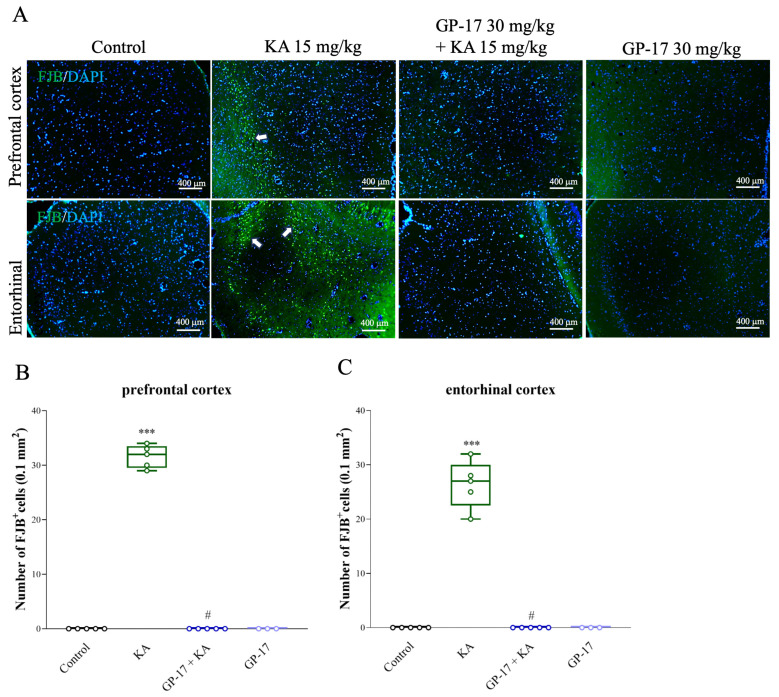
Effect of GP-17 on the neuronal degeneration in the entorhinal cortex of KA-treated rats. (**A**) FJB staining of 72 h after KA injection. FJB positive neurons are indicated by white arrows. Scale bars: 200 µm for 100×, 100 µm for 200×, 50 µm for 400×. (**B**,**C**). Statistical graph of FJB positive cells in the prefrontal cortex and entorhinal cortex in different groups. ***, *p* < 0.001 versus the control group, #, *p* < 0.05 versus the KA group. One-way ANOVA with Tukey posthoc test. *n* = 5 rats per group.

**Figure 4 biomolecules-14-00589-f004:**
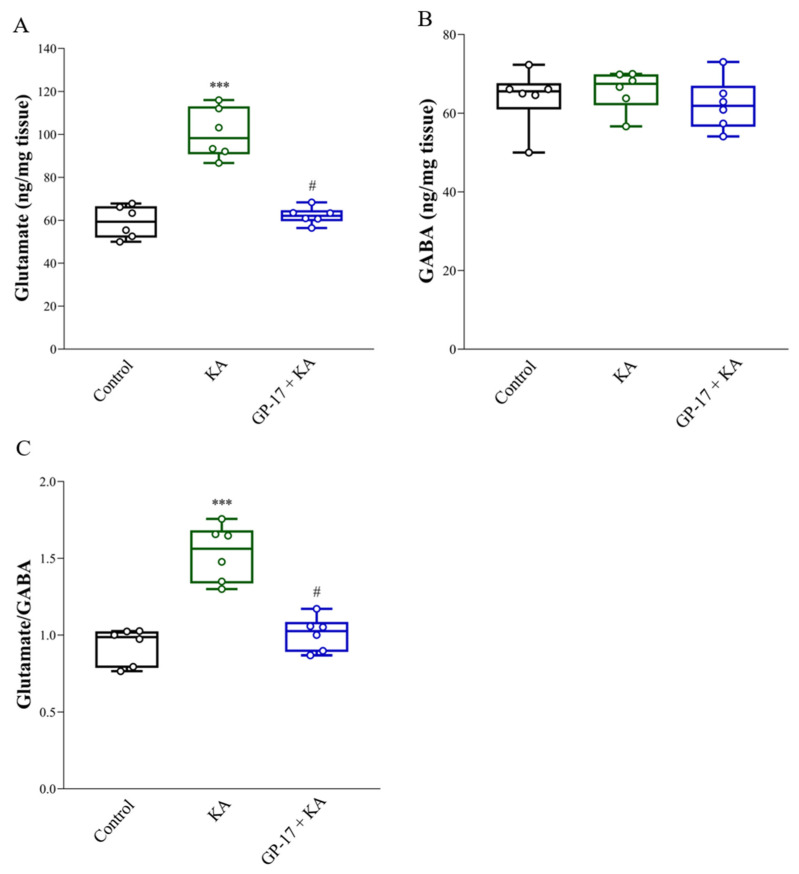
Effect of GP-17 on the concentration of glutamate (**A**) and GABA (**B**) and glutamate/GABA ratio (**C**) in the cortex from different groups. Glutamate and GABA levels were analyzed in the cortex of rats 72 h after KA injection. ***, *p* < 0.001 versus the control group, #, *p* < 0.05 versus the KA group. One-way ANOVA with Tukey posthoc test. *n* = 6 rats per group.

**Figure 5 biomolecules-14-00589-f005:**
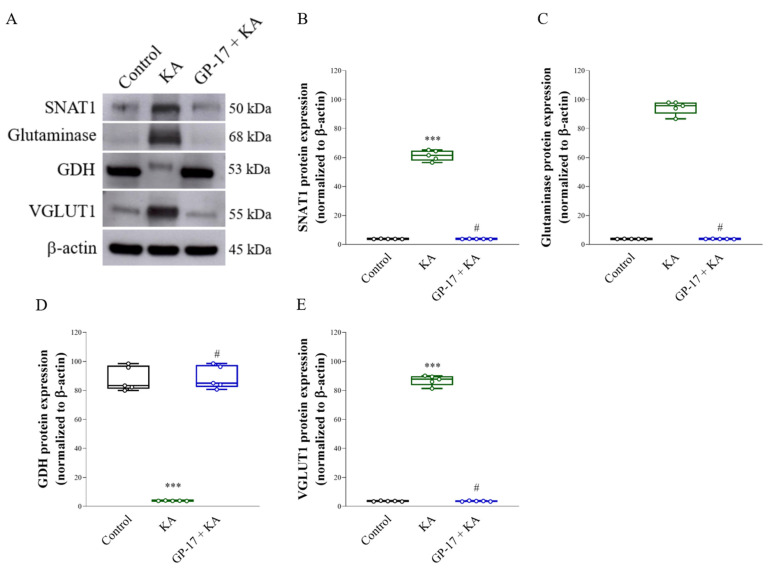
Effect of GP-17 on the protein expression levels of SNAT1, glutaminase, GDH, and VGLUT1 in the cortex from different groups. (**A**) Immunoblot results of SNAT1, glutaminase, GDH, and VGLUT1 in the cortex of rats 72 h after KA injection and the respective bar graphs. (**B**–**E**) The relative protein levels were quantified. ***, *p* < 0.001 versus the control group, #, *p* < 0.05 versus the KA group. One-way ANOVA with Tukey posthoc test. *n* = 5 rats per group. The details of the Western blot original images are in Appendix A.

**Figure 6 biomolecules-14-00589-f006:**
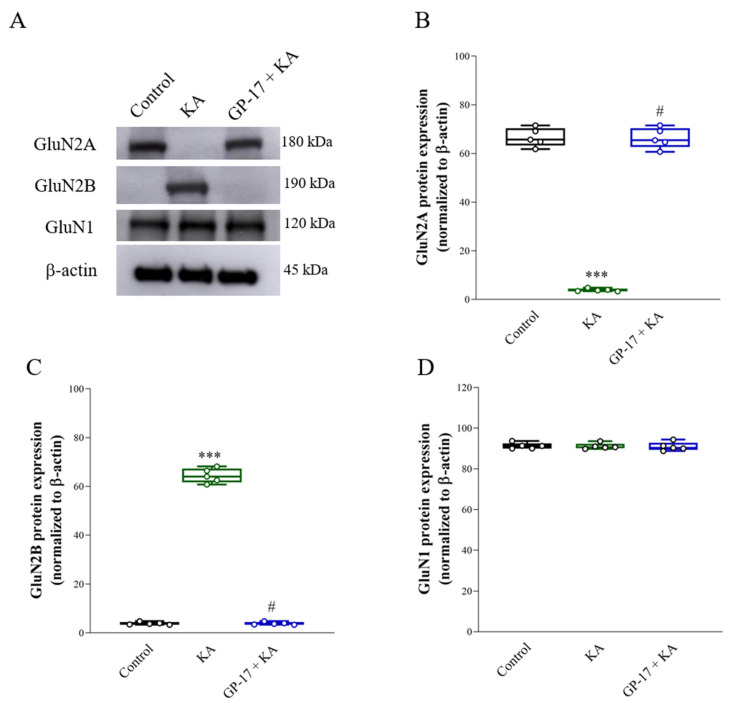
Effect of GP-17 on the protein expression levels of GluN2A and GluN2B in the cortex from different groups. (**A**) Immunoblot results of GluN2A, GluN2B, and GluN1 in the cortex of rats 72 h after KA injection. (**B**–**D**) The relative protein levels were quantified. ***, *p* < 0.001 versus the control group, #, *p* < 0.05 versus the KA group. One-way ANOVA with Tukey posthoc test. *n* = 5 rats per group. The details of the Western blot original images are in Appendix A.

**Figure 7 biomolecules-14-00589-f007:**
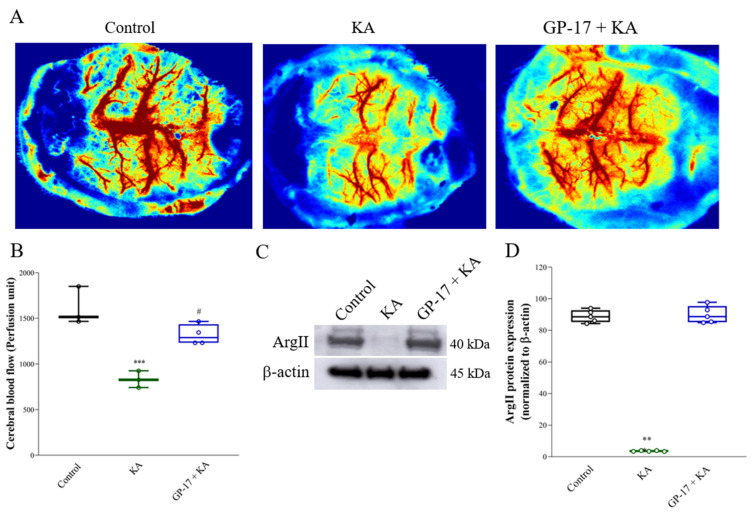
Effect of GP-17 on the cerebral blood flow in KA-treated rats. (**A**) The typical picture of cerebral blood flow in rats 72 h after KA injection. (**B**) The perfusion unit of cerebral blood flow in different groups. (**C**) Immunoblot results of GluN2A and GluN2B in the cortex of rats 72 h after KA injection. (**D**) The relative protein levels were quantified. **, *p* < 0.01 versus the control group, ***, *p* < 0.001 versus the control group. #, *p* < 0.05 versus the KA group. One-way ANOVA with Tukey posthoc test. *n* = 3–4 rats per group. The details of the Western blot original images are in Appendix A.

**Figure 8 biomolecules-14-00589-f008:**
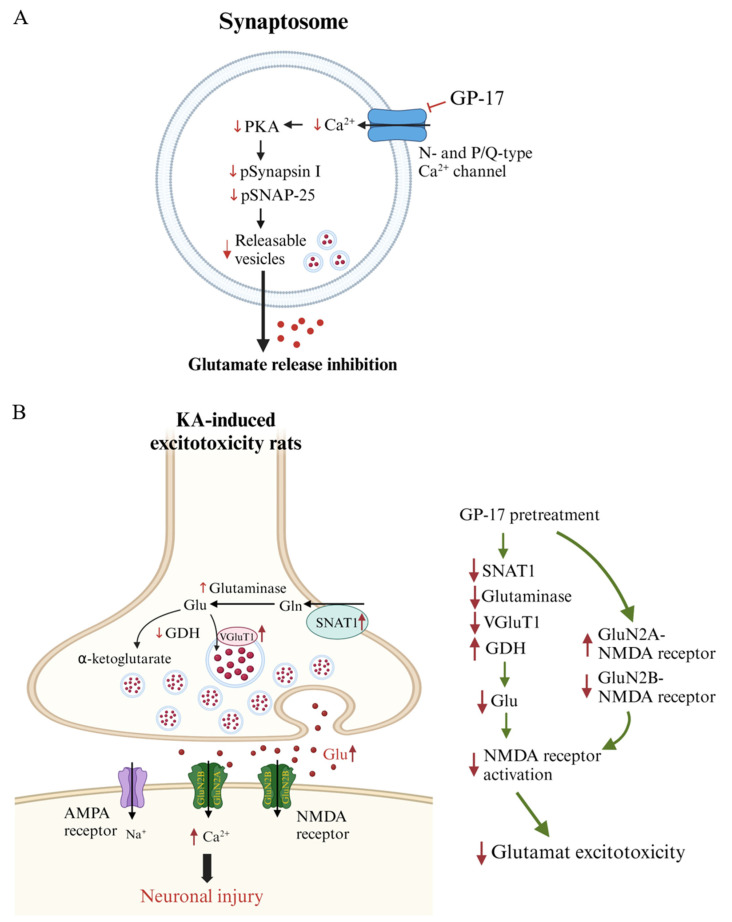
Schematic representation of GP-17-mediated inhibition of glutamate release from synapsosomes (**A**) and the neuroprotective effects oof GP-17 on KA-induced glutamate excitotoxicity in rats (**B**). Graphs created with BioRender.com accessed on 20 February 2024.

**Table 1 biomolecules-14-00589-t001:** Effect of GP-17 on KA-induced seizures in rats.

	KA 15 mg/kg	GP-17 10 mg/kg + KA 15 mg/kg	GP-17 30 mg/kg + KA 15 mg/kg
**Seizure score**	4.7 ± 0.1	4.3 ± 0.2	0.8 ± 0.3 ***
**Latency to first seizure (min)**	84.4 ± 6.5	111.1 ± 13.4	-
**% Seizure**	20/20 (100%)	9/10 (90%)	0/14 (0%)
**Mortality**	6/20	2/10	0/14

***, *p* < 0.001 versus the KA group. One-way ANOVA with Tukey posthoc test.

## Data Availability

The data presented in this study are available on request from the corresponding author.

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
