# Peer review of "Gypenoside XVII Reduces Synaptic Glutamate Release and Protects against Excitotoxic Injury in Rats"

_biomolecules, 2024, doi:10.3390/biom14050589_

Round 1
Reviewer 1 Report
Comments and Suggestions for Authors
In the paper entitled „Gypenoside XVII reduces synaptic glutamate release and protects against excitotoxic injury in rats” the authors using in vitro (cortical synaptosomes) as well in vivo model (KA-induced excitotoxicity) showed inhibitory effect of GP-17 on stimulated glutamate release mediated via the suppression of N- and P/Q-type Ca2+ channels and PKA inhibition, as well anti-epileptic and neuroprotective effects in animal model. The obtained results are new in respect to the effect of GP-17 in excitotoxicity models. In general study is well designed, various methods were engaged, data were well analyzed. There are some issues for explanation or correction for improvement before paper publication, as listed below:
1. Introduction: in paragraph describing the neuroprotective effects of Gypenoside XVII (GP-17) should be also mentioned work of Su et al. (Molecules 2023 Feb 27;28(5):2194. doi: 10.3390/molecules28052194) where this compound among other derivatives were tested for neuroprotection in SH-SY5Y cells exposed to glutamate. In this context the last sentence of Introduction “To the best of our knowledge, this study is the first to investigate the role of GP-17 in the glutamatergic system” should be modified. Also sentences in line 399 should be modified.
2. Materials and methods:
- In chapter 2.3 should be stated how many animals were used for synaptosomes preparation;
- In chapter 2.4 should be also described treatment with GP-17;
- In chapter 2.5 should be described solvents for GP-17 and KA, how control group of animals were treated, how many animals were tested in particular experimental group;
- For each used method in in vivo part should be mentioned number of animals used;
- In chapter 2.9 should be specified how particular phosphorylated and total proteins as well as protein load (beta-actin) were measured – on the same blots as measured proteins or on sister ones, on original or stripped blots?
3. Results:
- In chapter 3.1. should be justified the chosen concentrations of GP-17, as well other compounds;
- In Fig. 1 legend wrong described panels, not stated chemical structure of GP-17;
- Fig. 2, Fig. 5, Fig. 6 – results from WB should be demonstrated in separate graphs, as were analyzed by one-way ANOVA (separate graph for each protein);
- Fig. 3 – data on histograms should be shown on separate graphs, as were analyzed by statistical test;
- Lack of Table 1;
- Fig. 6 – there should be also measured NR1 subunit, since it is crucial for functional NMDA receptors;
- In Supplementary materials should be indicated experimental groups near representative blots, as well as MW.
4. Discussion:
- In third paragraph there should be at least some discussion regarding synergism between PKC inhibitor and GP-17 in attenuation of glutamate release;
- There should be discussion of obtained data with GP-17 to the effectiveness to other plant derived compounds with similar chemical structure, their neuroprotective potency should be compared (if such data are available).
5. Minor remarks:
- Lines 58/59 – two times written about neuroprotective effects;
- Line 202 – check units for protein load on gel;
- Line 405 – check units, seems to be wrong (IC50 of 16 mM).
Author Response
Reviewer 1
We thank the reviewer for the critical comments and constructive suggestions.
- Introduction: in paragraph describing the neuroprotective effects of Gypenoside XVII (GP-17) should be also mentioned work of Su et al. (Molecules 2023 Feb 27;28(5):2194. doi: 10.3390/molecules28052194) where this compound among other derivatives were tested for neuroprotection in SH-SY5Y cells exposed to glutamate. In this context the last sentence of Introduction “To the best of our knowledge, this study is the first to investigate the role of GP-17 in the glutamatergic system” should be modified. Also sentences in line 399 should be modified.
The sentences are modified (Page 2, lines 57-58) and deleted (Page 2, lines 74-75).
- Materials and methods:
- In chapter 2.3 should be stated how many animals were used for synaptosomes preparation;
The number of animals is added (Page 3, line 96).
- In chapter 2.4 should be also described treatment with GP-17;
The sentence "GP-17 was added to synaptosomes 10 min before 4-AP. Other additions before 4-AP were made as detailed in the figure legends² is added (page 3, lines 119-120).
- In chapter 2.5 should be described solvents for GP-17 and KA, how control group of animals were treated, how many animals were tested in particular experimental group;
The sentences ²GP-17 (30 mg/kg) was dissolved in a saline solution containing 1% DMSO and was administered (i.p.) 30 min before KA (15 mg/kg in 0.9% NaCl, pH 7.0, i.p.) injection." is added (page 3, lines 143-145).
- For each used method in in vivo part should be mentioned number of animals used;
The number of animals is added (Page 3, line 146; Page 4, line 147, 157, 184, 195; Page 5, line 205).
- In chapter 2.9 should be specified how particular phosphorylated and total proteins as well as protein load (beta-actin) were measured – on the same blots as measured proteins or on sister ones, on original or stripped blots?
The sentence "The measurements of b-actin were conducted on sister blots, with each blot loaded with 30 mg of protein per lane. ² is added (Page 5, lines 232-234).
- Results:
- In chapter 3.1. should be justified the chosen concentrations of GP-17, as well other compounds;
The sentences ²Given the robust depression of glutamate release seen with 10 μM GP-17, this concentra-tion was used in subsequent experiments to evaluate the mechanisms underlying the ability of GP-17 to reduce glutamate release." and "The dose of administration of ω-CgTX GVIA and ω-AgTX IVA were chosen based on previous studies [25,26] " are added (Page 6, 251-253, 273-274).
- In Fig. 1 legend wrong described panels, not stated chemical structure of GP-17;
The sentence "Chemical structure of GP-17.² is added (Page 7, line 277).
- Fig. 2, Fig. 5, Fig. 6 – results from WB should be demonstrated in separate graphs, as were analyzed by one-way ANOVA (separate graph for each protein);
Fig. 2, Fig. 5, and Fig. 6 are modified.
- Fig. 3 – data on histograms should be shown on separate graphs, as were analyzed by statistical test;
Fig. 3 is modified.
- Lack of Table 1;
Table 1 is added.
- Fig. 6 – there should be also measured NR1 subunit, since it is crucial for functional NMDA receptors;
The protein expression level of GluN1 is examined. The result is included in Fig. 6.
- In Supplementary materials should be indicated experimental groups near representative blots, as well as MW.
The experimental groups and MW are included in supplementary materials.
- Discussion:
-In third paragraph there should be at least some discussion regarding synergism between PKC inhibitor and GP-17 in attenuation of glutamate release;
The sentences² In addition, we found that the inhibitory effect of GP-17 on 4-AP-evoked glutamate release was not affected by the PKC inhibitor GF109203X. In the presence of GF109203X, GP-17 still induced a 49% decrease in the glutamate release. The additive relationship between GF109203X and GP-17 indicates that PKC is not involved in the effect of GP-17 on gluta-mate release." are added in the discussion (Page 21, lines 467-471).
- There should be discussion of obtained data with GP-17 to the effectiveness to other plant derived compounds with similar chemical structure, their neuroprotective potency should be compared (if such data are available).
The sentences "In addition, the dose of GP-17 used to prevent neuronal damage in our in vivo study was 30 mg/kg. Consistent with our study, studies have revealed that the effect of GP-17 occurs at 10–50 mg/kg in vivo. Other gypenosides administered at doses of 100-200 mg/kg can attenuate depression- or anxiety-like behavior of the mice [69] and improve memory impairment in rats.² are added in discussion (Page 22, lines 551-556).
- Minor remarks:
- Lines 58/59 – two times written about neuroprotective effects;
It is correct.
- Line 202 – check units for protein load on gel;
Checked (line 233).
- Line 405 – check units, seems to be wrong (IC50 of 16 mM).
Checked (line 443).
Reviewer 2 Report
Comments and Suggestions for Authors
The study conducted by Lu and coworkers describes the mechanism of how GP-17 inhibits glutamate excitotoxicity. Overall, the study is well designed and the conclusions are supported by the results. I have some minor comments:
line 55-59 I would indicate that these effects are observed in various in vitro and in vivo models (it is not a standard therapy in humans).
why did the authors use 6-week- old rats? They are periadolescent, normally slightly older rats are used as adults..
I am worried about cervical dislocation. 180-200 g is quite a lot for this kind of sacrifice.
Did the Authors consider including a GP-17-only group? To see the effect of this compound on control animals.
line 105 ' 0.5 mg of synaptosomal suspension was diluted in HEPES buffer medium and centrifuged at 3000× g for 10 min at 4 °C. T' Is this done after determining the concentration? What is the purpose of this step?
Please include all important parameters when reporting ANOVA results (F=...etc)
'Equal aliquots of protein (20 g/lane) from each sample were separated via SDS‒PAGE.' Is it 'mg' or 'g'?
Author Response
Reviewer 2
We thank the reviewer for the critical comments and constructive suggestions.
The study conducted by Lu and coworkers describes the mechanism of how GP-17 inhibits glutamate excitotoxicity. Overall, the study is well designed and the conclusions are supported by the results. I have some minor comments:
line 55-59 I would indicate that these effects are observed in various in vitro and in vivo models (it is not a standard therapy in humans).
The sentence is modified to ²Several experimental studies have reported various pharmacological properties to GP-17, including immunoregulatory, anti-inflammatory, neuroprotective, cardioprotective, lipid-lowering, and neuroprotective effects" (Page 2, lines 57-59).
why did the authors use 6-week- old rats? They are periadolescent, normally slightly older rats are used as adults.
The method of animal sacrifice is cervical dislocation, and the weight of the animal must be less than 200g according to the animal care and use regulations.
I am worried about cervical dislocation. 180-200 g is quite a lot for this kind of sacrifice.
The method of animal sacrifice is cervical dislocation, and the weight of the animal must be less than 200g according to the animal care and use regulations.
Did the Authors consider including a GP-17-only group? To see the effect of this compound on control animals.
The result of GP-17-only group is added in Fig.3.
line 105 ' 0.5 mg of synaptosomal suspension was diluted in HEPES buffer medium and centrifuged at 3000× g for 10 min at 4 °C. T' Is this done after determining the concentration? What is the purpose of this step?
Th sentence is modified to ²The supernatants were discarded, and the synaptosomal pellets were stored on ice and used for glutamate release, FM1-43 release and Western blot within 4–6 h.² (Page 3, line 109-111).
Please include all important parameters when reporting ANOVA results (F=...etc)
The values of F are included in result section.
'Equal aliquots of protein (20 g/lane) from each sample were separated via SDS‒PAGE.' Is it 'mg' or 'g'?
It is mg (Page 5, line 211).
Round 2
Reviewer 1 Report
Comments and Suggestions for Authors
In the revised paper authors answered sufficiently for arisen issues and made relevant improvements in the revised manuscript. There are only some minor points for clarification before paper publication:
- Lines 148/149 – in the sentence “The rats were divided into the following three groups: control group (i.p. injection of DMSO; n = 10)” should be added what percentage of DMSO was given to control group, the same as is mentioned in solvent for GP-17? Has been control group administered with vehicle for KA? It is not clear if KA group received also an vehicle for GP-17?
- There is some inconsistency between number of animals mentioned in Materials and methods concerning HPLC method (n=5), whereas when described results on Fig. 4 there is n=6. It should be clarified.
- In Fig. 3 was added data for GP-17 group. However this was not mentioned in Materials and methods, when authors described Experimental design. For KA model. It should be checked and corrected.
Author Response
We thank the reviewer for the critical comments and constructive suggestions.
In the revised paper authors answered sufficiently for arisen issues and made relevant improvements in the revised manuscript. There are only some minor points for clarification before paper publication:
- Lines 148/149 – in the sentence “The rats were divided into the following three groups: control group (i.p. injection of DMSO; n = 10)” should be added what percentage of DMSO was given to control group, the same as is mentioned in solvent for GP-17? Has been control group administered with vehicle for KA? It is not clear if KA group received also an vehicle for GP-17?
The sentence is modified (Page 3, lines 143-144).
- There is some inconsistency between number of animals mentioned in Materials and methods concerning HPLC method (n=5), whereas when described results on Fig. 4 there is n=6. It should be clarified.
The number is changed to 6 (Page 4, line 181).
- In Fig. 3 was added data for GP-17 group. However this was not mentioned in Materials and methods, when authors described Experimental design. For KA model. It should be checked and corrected.
The sentence is modified (Page 3, lines 145-146).